# Plasticity-Related Activity in the Hippocampus, Anterior Cingulate, Orbitofrontal, and Prefrontal Cortex Following a Repeated Treatment with D_2_/D_3_ Agonist Quinpirole

**DOI:** 10.3390/biom11010084

**Published:** 2021-01-11

**Authors:** Hana Brozka, Daniela Alexova, Dominika Radostova, Martina Janikova, Branislav Krajcovic, Štěpán Kubík, Jan Svoboda, Ales Stuchlik

**Affiliations:** 1Laboratory of the Neurophysiology of Memory, Institute of Physiology of the Czech Academy of Sciences, Videnska 1083, 142 20 Prague, Czech Republic; daniela.alexova@fgu.cas.cz (D.A.); dominika.radostova@fgu.cas.cz (D.R.); martina.janikova@fgu.cas.cz (M.J.); branislav.krajcovic@fgu.cas.cz (B.K.); stepan.kubik@fgu.cas.cz (Š.K.); svobodaj@biomed.cas.cz (J.S.); 2Second Faculty of Medicine, Charles University, 142 20 Prague, Czech Republic; 3First Faculty of Medicine, Charles University, 142 20 Prague, Czech Republic; 4Third Faculty of Medicine, Charles University, 142 20 Prague, Czech Republic

**Keywords:** hippocampus, obsessive-compulsive disorder, quinpirole, stereotypical checking, *Arc*, *Homer1*

## Abstract

Quinpirole (QNP) sensitization is a well-established model of stereotypical checking relevant to obsessive-compulsive disorder. Previously, we found that QNP-treated rats display deficits in hippocampus-dependent tasks. The present study explores the expression of immediate early genes (IEG) during QNP-induced stereotypical checking in the hippocampus, anterior cingulate cortex (ACC), orbitofrontal cortex (OFC), and medial prefrontal cortex (mPFC). Adult male rats were injected with QNP (0.5 mg/mL/kg; *n* = 15) or saline (*n* = 14) daily for 10 days and exposed to an arena enriched with two objects. Visits to the objects and the corners of the arena were recorded. QNP-treated rats developed an idiosyncratic pattern of visits that persisted across experimental days. On day 11, rats were exposed to the arena twice for 5 min and sacrificed. The expression of IEGs *Arc* and *Homer1a* was determined using cellular compartment analysis of temporal activity by fluorescence in situ hybridization. IEG-positive nuclei were counted in the CA1 area of the hippocampus, ACC, OFC, and mPFC. We found significantly fewer IEG-positive nuclei in the CA1 in QNP-treated rats compared to controls. The overlap between IEG expressing neurons was comparable between the groups. We did not observe significant differences in IEG expression between QNP treated and control rats in ACC, OFC, and mPFC. In conclusion, treatment of rats with quinpirole decreases plasticity-related activity in the hippocampus during stereotypical checking.

## 1. Introduction

Obsessive-compulsive disorder (OCD) is characterized by intrusive thoughts followed by repetitive physical and/or mental acts. Currently, OCD is one of the leading causes of disability worldwide [1] with an immense impact on the well-being of patients and their relatives [2]. Fortunately, in past decades, our understanding of OCD has greatly advanced, allowing the development of several pharmacological and non-pharmacological therapies of OCD [3,4]. Yet, as many as 80% of OCD patients do not experience a full long-term remission of symptoms following therapy [5], which indicates that our understanding of OCD is still limited.

Cortico-striato-thalamo-cortical circuits (CSTC) are widely considered to be the neuroanatomical substrates of OCD. Hyperactivity within these circuits has been implicated repeatedly in OCD by neuroimaging studies. In fact, the hyperactivity of the caudate nucleus, the orbitofrontal cortex (OFC) and the anterior cingulate cortex (ACC) is the most consistently replicated finding in OCD [6,7,8]. However, abnormalities in several brain regions outside CSTC circuits have also been reported in OCD, but these brain regions have received little scientific attention in terms of OCD etiology [9,10,11]. One of these previously disregarded brain regions is the hippocampus.

The idea that the hippocampus is involved in OCD gained traction in recent years. Several studies showed functional and morphological changes of the hippocampus in OCD patients. Functional imaging studies reported hippocampal hyperactivity in OCD patients during symptom provocation resulting in obsessive thoughts [12] and during performance of cognitive tasks [13,14] Kang et al. showed that the increased activity of patients’ hippocampi normalized following successful SSRI treatment [15]. Atmaca et al. observed that in hippocampal N-acetyl-l-aspartate/choline ratio is reduced in OCD patients, which indicates ongoing neurodegenerative changes [16]. Several small-scale studies reported reductions of the hippocampal volume in OCD patients [15,17,18]; findings later confirmed by ENIGMA study—a worldwide collaborative initiative to elucidate the neuronal substrates of OCD [19].

Several case studies suggest that hippocampal damage can increase the chance of developing OCD. OCD was detected in temporal lobe epilepsy patients [20,21] and following traumatic brain injury to the temporal lobe region [22]. In a review, Isaac et al. showed that patients with temporal lobe epilepsy had a high incidence of comorbid OCD [23]. Interestingly, temporal lobe epilepsy patients displayed frequent compulsive behavior, while obsessive thoughts were generally absent. These studies indicate that the neurobiological pathways sub-serving compulsions may differ from those underlying obsessive thoughts [24]. In summary, human studies indicate that increased hippocampal activity is associated with obsessive thoughts while decreased hippocampal activity is associated with compulsive behaviors.

Neuronal substrates of many aspects of psychiatric disorders can be probed using animal models. In the case of OCD, employing animal models to study obsessive thoughts in animals is at present impossible. However, animal models can be utilized to study neuronal processes associated with compulsive and stereotypical behavior. Stereotypical behavior can be assessed by diverse behavioral protocols ranging from spontaneous alternation and compulsive lever pressing to spontaneous stereotypical checking. Spontaneous stereotypical checking induced by a D_2_/D_3_ agonist quinpirole (QNP) resembles the compulsive behavior of OCD patients [25]. Previously, we reported that QNP treated rats display a profound, yet transient, deficit in the hippocampus-dependent active place avoidance task [26,27]. In the present study, we examined if QNP-induced stereotypical checking alters expression of plasticity-inducing IEG expression in the hippocampus and cortical regions implied in OCD. To our best knowledge, there are no other studies looking at the role of hippocampus in etiology of OCD-like behavior.

## 2. Materials and Methods

First, we induced stereotypic checking behavior in rats by repeated administration of QNP followed by exposure to an enriched open field arena similar to one described previously [25]. Next, after development of stereotypic behavior in the QNP treated rats, we used two 5 min exposures to the same arena separated by a 20 min interval to induce the expression of two IEGs: *Homer1a* and *Arc*. IEGs expression was determined using cellular compartment analysis of temporal activity by fluorescence in situ hybridization (catFISH) in several brain regions: ACC, OFC, and CA1 area of the hippocampus, and the medial prefrontal cortex (mPFC). ACC and OFC are implicated in OCD pathophysiology. CA1 area was selected because most of the IEG expression occurs in the CA1 area: Approximately 30–40% of CA1 neurons express the IEG *Arc* after behavioral exploration of (novel) environment. In comparison, only 10–15% of CA3 neurons and 2% dentate granule cells express *Arc* in the same conditions [28]. Therefore, the majority of the active neurons in the hippocampus are from CA1. For this reason, most studies on IEG expression in the hippocampus focus on CA1. Furthermore, expression of *Arc* in the DG is not short-lasting such as in CA1, CA3, and all other brain regions, but once triggered, remains high for as long as 8 h [29,30]. mPFC was selected as an accessible control region with no reported connection to OCD pathophysiology.

### 2.1. Animals

Twenty-nine adult male Long–Evans rats from the breeding colony of the Institute of Physiology (derived from Charles River breeding core) were used. At the start of the experiment, rats weighed 300–350 g and were 12–13 weeks old. The rats were housed in groups of two to three per cage in an air-conditioned rat room with a stable temperature of 22 °C, and 12/12-light/dark cycle. All experiments were conducted in the light phase of the day. Food and water were available ad libitum. Before the experiments, rats were handled for 2 min daily for three days. All rat manipulations were approved by a local ethical committee and complied with the Animal Protection Code of the Czech Republic and with a corresponding directive of the European Community Council on the use of laboratory animals (2010/63/EC).

### 2.2. Drug Administration

Rats were divided into QNP treatment (*n* = 15) and control (*n* = 14) groups. The treatment group received one daily subcutaneous injection of QNP dissolved in saline at a dose of 0.5 mg/mL/kg (Q102; Sigma Aldrich, Prague, Czech Republic); the control group received one daily subcutaneous injection of saline of 1 mL/kg. Following drug administration, animals were left undisturbed in their home cages for 50 min and then individually tested in the object-enriched arena.

### 2.3. The Apparatus: Object-Enriched Arena

The arena consisted of a square box with a white plastic floor (95 × 95 cm^2^) surrounded by 50 cm tall walls made of black non-reflective plastic. Two metal objects were fixed to arbitrary locations in the arena, where they remained throughout the experiment. The arena was elevated 1 m above the room floor and a video camera was mounted above the center of the arena. The camera was connected to a computer located in an adjacent room, which allowed the experimenter to observe and record each rat’s behavior simultaneously.

### 2.4. Behavioral Procedure

We monitored the spontaneous behavior of rats during repeated QNP and control treatment during 10 daily 50 min exposures to the object-enriched arena. Each experimental day began by placing a rat in the corner of the arena, facing the wall. Between the rats, the arena was cleaned with 2.5% acetic acid. The video recording of the rat’s spontaneous behavior started the moment the experimenter left the room. The rat was left undisturbed to explore the arena freely for 50 min, and then it was returned to its home cage.

The course of the experimental day 11 was modified to optimize the induction of IEG expression by the behavior of interest. Twelve hours before testing, all rats were single-housed in white opaque home cages to minimize baseline IEG expression and enhance the signal/noise ratio by sensory restriction. Randomly selected QNP treated (*n* = 9) and control (*n* = 9) rats explored the arena (A_1_A_2_) on the experimental day 11; the remaining QNP treated (*n* = 6) and control rats (*n* = 5) were left undisturbed in their opaque home cages (HC) for the duration of the experiment. Arena-exploring rats were placed into the arena for 5 min (session A_1_), returned to the home cage for 20 min, and then placed in the arena again for another 5 min (session A_2_). Immediately after the end of session A_2_ or 80 min after the injection in the HC groups, the rats were deeply anesthetized with 2% isoflurane and decapitated. Their brains were quickly removed (<2 min), flash-frozen in a dry-ice-cooled isopentane bath, and stored in a freezer at −80 °C. The experimental scheme is illustrated in Figure 1A.

### 2.5. Behavioral Analysis and Statistics

The 10 recordings of the 50 min long exposures to the enriched arena were analyzed using a video-tracking system to detect and track the position of each rat’s head, body, and tail (Viewer2, Biobserve GmbH, Bonn, Germany). We focused the analysis on the locomotor activity and the most salient spatial locations for compulsive checking: The two objects and the four corners of the arena. Software parameters were optimized based on the manual scoring of several random tracks. The main output parameter was the number of visits to each of these locations. Determination of the optimal location size was important, as entry into the location was counted only when an animal entered the location and interacted with the object/corner. On the other hand, location entry was not counted when a rat was merely passing by. To determine the optimal size and shape of the location containing the objects and the cornets, we used an empirical approach. To that aim, we utilized data from pilot experiments. The optimal location was either circular (objects) or quadri-circular (corners). The radius of locations was determined by comparing the scoring of the software with handmade scores of visits by an experimenter blinded to the treatment condition. We selected a final radius of each location so that experimenter-determined counts of a visit to objects/corners corresponded exactly to the counts of visits by the Biobserve Viewer2 software. In our setup, object zones had a radius of 10 cm with the object in the center, while quadri-circular corner zones had a radius of 15 cm with the tip in the corner of the arena. The radius of the zones would be different if objects were of different sizes or if animals were bigger/smaller. After we obtained the number of visits to spatial locations, we summed up and ranked visits to each location on days 1–10 separately for each rat. We ranked locations from the most to least visited and labeled these locations zones A–F: The most visited location was labeled “zone A” and the least visited location was labeled “zone F”. This allowed us to pool the data from animals that displayed different spatial patterns of activity.

### 2.6. Tissue Preparation

We processed the brain tissue to evaluate IEG expression in several brain regions. Note that both IEGs are triggered by both sessions, but the intranuclear signal from intron-enriched probes for *Arc* triggered by the first session has dissipated by the time of sacrifice, while the signal from the probes targeting distal regions of the long primary transcript of *Homer1a* has not yet appeared at the same moment. Thus, *Homer1a* probes detect plasticity-inducing activity during the first session, while *Arc* probes mark the activity-inducing activity during the second exploration session. The *Arc*/*Homer1a* FISH procedure was previously described in detail elsewhere [31,32]. Briefly, frozen 4 mm coronal segments from the right hemispheres containing the dorsal hippocampus were arranged in blocks of 6–8 samples to ensure simultaneous processing and maximize the number of between-group comparisons on each slide. The blocks were embedded in Optimal Cutting Temperature medium (OCT; Sakura Finetek Europe, Alphen aan den Rijn, Netherlands), 20 µm coronal sections were cut in a cryostat (Leica CM 1850, Wetzlar, Germany), and mounted on gelatin-coated Superfrost slides (Fisher Scientific, Pardubice, Czech Republic).

The best-quality sections of each block were processed for fluorescence in situ hybridization to determine IEG expression during two 5 min sessions of day 11. Three slides across the rostro-caudal axis were selected to provide a representative sample for each evaluated brain region (details in the “Image analysis”). Samples (slide-mounted brain sections) were fixed in an ice-cooled 4% solution of buffered paraformaldehyde, then incubated in a mixture of 0.5% acetic anhydride with 1.5% triethanolamine, and permeabilized in a 1:1 acetone/methanol mix at −20 °C. Samples were immersed in a hybridization buffer containing hapten-labeled antisense RNA probes for *Arc* intron sequences and *Homer1a* 3′UTR (untranslated region), coverslipped, and incubated overnight at 56 °C. To degrade unbound RNA, we treated the samples with 0.1 μg/mL RNAse A in a citrate buffer. Endogenous peroxidase activity was quenched with 3% H_2_O_2_. The probes were detected serially using anti-hapten antibodies conjugated with horseradish peroxidase (Jackson ImmunoResearch Europe, Ely, UK; Roche, Prague, Czech Republic). The signal was then visualized using a tyramide signal amplification system (TSA kit, PerkinElmer, Waltham, MA, USA) with Cy3 and FITC fluorophores. Cellular nuclei were counterstained with DAPI (Invitrogen), and the slides were cover-slipped with anti-fade medium (Vectashield, Vector Laboratories, Burlingame, CA, USA) and sealed with nail polish.

### 2.7. Image Acquisition

Confocal stacks were acquired from the CA1 region of the hippocampus, ACC, OFC, and mPFC on a Leica TCS SP8 laser-scanning microscope with an apochromatic HCX PL APO 20× immersion objective. Each stack was composed of 21 optical sections ~1 µm apart. The blue nuclear counterstain by DAPI was imaged using 405 nm excitation and a 415–490 nm emission bandpass. The green signal from *Homer1a* probes (TSA-FITC) was excited with a 488 nm laser and imaged within a 510–550 nm emission bandpass. The orange/red signal from *Arc* probes (TSA-Cy3) was excited at 555 nm and filtered using a 565–665 nm emission bandpass. The laser power, gain, and offset were always set for an entire slide. The settings were optimized for the detection of bright intra-nuclear foci of *Homer1a* and *Arc* expression, marking ongoing nuclear transcription.

### 2.8. Image Analysis

Six to nine images from CA1 and two to four images from ACC, OFC, and mPFC were analyzed from each animal. The analysis was conducted using a custom macro running in Fiji software [33] by an experimenter blinded to the samples’ experimental identity. Only neuronal nuclei in the middle 20% of the Z-stack were selected for analysis to avoid the inclusion of fragmented cell nuclei from the section edge yielding false-negative results (due to a cut-off signal). Putative glial cell nuclei, identified by their smaller size, prolonged shape, and bright, unstructured nuclear counterstain that showed no IEG signal were excluded from the analysis.

Neuronal nuclei were automatically classified as negative, *Arc*-positive, *Homer1a*-positive, or double-positive, relative to threshold values based on manually selected nuclei with the weakest signal observed in the exploring rats. Home cage controls rats were not used in the threshold selection due to their low expression levels and insufficient numbers of nuclei with weak expression. Threshold values were integrated across each slide, but separately for each region and channel (green and orange/red). The thresholded IEG signals were converted to binary variables and the numbers and proportions of nuclei positive for either one or both IEGs were calculated. This methodology enabled us to assess the activity patterns of hundreds of neurons during two distinct periods in multiple brain regions per animal.

### 2.9. Similarity Score—A Measure of IEG Overlap between A_1_ and A_2_

Next, we tested if the QNP induced stereotyped behavior is reflected in increased recruitment of the same IEG expressing neurons. To examine changes in the activity patterns between the first and second exploration session, we calculated the similarity score. This parameter reduces four variables (proportions of negative, *Arc*+, *Homer1a*+, and double+ neuronal nuclei) into a single value normalized to the inter-individual variability in the IEG expressing neuronal population size. The evaluated brain regions included the hippocampal CA1, ACC, OFC, and mPFC. Similarity scores were calculated as described earlier [31,34]. For an exact calculation of similarity score, we refer an interested reader to these publications.

### 2.10. Statistical Analysis

The data were statistically analyzed using general mixed and linear models. Parametric assumptions were tested using the Shapiro–Wilks’ test and Levene’s test. In the present case, data met parametric assumptions; they were not transformed. To analyze behavioral tests, we used general mixed models (GMM). In the case of significant GMM results, GMMs were followed by separate ANOVAs or repeated measure ANOVAs. In the case of IEGs expression, two-way ANOVA was used to analyze each brain region separately. Significant ANOVA results were followed with either the Bonferroni post hoc test or a simple effect analysis to examine the between-group differences. All statistical analyses were conducted using SPSS software.

## 3. Results

### 3.1. Locomotor Sensitization Induced by QNP Treatment

We evaluated the locomotor activity of both groups of animals using two-way ANOVA with repeated measures and found a significant interaction between the treatment group (QNP or control) and the experimental day (*p* < 0.001). We found that the locomotor activity of the QNP group increased significantly with each experimental day (*p* < 0.05). The locomotor activity of the control rats did not differ significantly among the experimental days (*p* = 0.73): On the experimental day 10, the track length of the control rats was only 1.3 times longer than on the experimental day 1. The locomotor activity of the QNP and control groups on each experimental day was compared using simple effects analysis. The locomotion of the QNP rats was significantly higher than the locomotion of the control rats on all experimental days from the experimental day 4 to 10 (all *p* < 0.001). The average locomotion during experimental days 1–10 is shown in Figure 1B.

### 3.2. Stereotypical Checking—Experimental Days 1–10

Next, we compared the number of visits (dependent variable) among zones (A–F), treatments (QNP/control), and days (1–10) using GMM. We found a significant 3-way treatment–zone–day interaction (*p* < 0.001). In QNP treated rats, we found that zone A was visited significantly more than any other zone on the day 1 and on the days 4 to 10 (*p* < 0.001). On the day 10, zone B was more visited than zones C to F (*p* < 0.05). In the control group, there were no significant differences in the number of visits among zones on any experimental day. Visits to zones are visualized in Figure 1C–E.

Moreover, there were differences in the identity of the most frequented zones between the groups. In QNP treated rats the most visited zone, zone A, was most often a zone containing a corner; similarly, in controls zone A was also most frequently a zone containing a corner. On the other hand, in QNP treated rats the second most frequented zone, Zone B, was a zone containing an object; while in control rats, it was most often a zone containing a corner. To summarize, QNP treated rat stereotypic activity comprised most often of movement between the selected corner and the selected object, while control animal moved between corners of the arena.

### 3.3. Stereotypical Checking in Sessions A_1_ and A_2_ (Experimental Day 11)

We compared visits to zones in sessions A_1_ and A_2_ (both 5 min) of experimental day 11 with the visits in experimental day 10 (50 min). First, we standardized the data from these sessions to be able to directly compare checking between the shorter A_1_ and A_2_ sessions and the longer experimental day 10. We analyzed differences in the number of visits among treatments, zones, and sessions (day 10, A_1_, A_2_) using GMM. The GMM statistical test was significant (*p* < 0.05) and was followed by a separate ANOVA for each session and each treatment. In the A_1_ session, QNP treated rats showed significantly more visits to zone A compared to zones E and F. In the A_2_ session, QNP treated rats made significantly more visits to zone A compared to zones D (*p* = 0.008), E, and F (*p* < 0.001). Here we showed that QNP treated rats maintained the preference of zone A compared with zones E and F from day 10 to A_1_ and A_2_ sessions of day 11 (Figure 2A). In contrast, control rats did not prefer any zone significantly more in any session (Figure 2B).

Because IEG expression can reflect an animal’s movement through the environment, we compared the locomotion activity of QNP and control rats during sessions A_1_ and A_2_. In session A_1_, we found no statistically significant difference in locomotion between QNP and control rats (Figure 2C; *p* = 0.186). In session A_2_, QNP rats displayed significantly higher locomotion than control rats (Figure 2D; *p* = 0.025).

### 3.4. IEG Expression in the Hippocampus, OFC, ACC, and mPFC

We used three-way ANOVA with repeated measures on the proportions of *Arc*+ and *Homer1a*+ nuclei. IEG expression in the hippocampus, ACC, OFC, and mPFC was analyzed separately (Figure 3). Independent variables were treatment (QNP or vehicle) and environment (A_1_A_2_ or HC), with repeated measures on the exploration session (A_1_ or A_2_). Highly significant effects of the environment showed that exploration triggered a marked increase in IEG expression in the hippocampus (F_1,23_ = 9.099; *p* < 0.006) and OFC (F_1,23_ = 9.598; *p* < 0.005) of exploring rats (A_1_A_2_) compared to rats remaining in their home cages (HC). However, in OFC there was no difference in IEG expression between QNP treated and control rats. In hippocampus, significant effect of treatment (F_1,23_ = 5.873; *p* = 0.024) indicating that QNP treated rats displayed reduced numbers of IEG-positive neuronal nuclei in exploring rats (A_1_A_2_) relative to vehicle controls (there were 29.4% *Arc*+ neurons in control rats, while there were only 11.2% *Arc*+ neurons in QNP treated rats in A_1_; and there were 30.6% *Homer1a*+ neurons in control rats, while there were 20.7% *Homer1a*+ in QNP treated rats in A_2_; despite the differences treatment x exploration session was not significant). In the HC, IEG expression was comparable between QNP treated and control rats. We found no statistically significant effects of environment, treatment, or session in the ACC and mPFC. In summary, QNP significantly decreases IEG expression specifically in CA1 region of hippocampus. The numbers of IEG expressing neuronal nuclei are visualized in Figure 2E.

### 3.5. Overlap of IEG Expressing Neurons between A_1_ and A_2_

Next, we assessed the overlap between populations of IEG expressing neurons in the same environment during two exposures to the same environment (A_1_ and A_2_) using similarity score. We used two-way ANOVA to compare similarity scores of *Homer1a* and *Arc* expressing neuronal nuclei between treatment and environment for each brain region separately (HPC, ACC, OFC, mPFC). Neither ANOVA was significant; indicating there is no difference between recruitment of neurons between QNP treated and control rats. In conclusion, QNP treated rats did not show the expected increased overlap in IEG expressing neurons between sessions A_1_ and A_2_ in any of the explored regions compared to control rats. The similarity scores for each brain region are visualized in Figure 2E(c,f,i,l).

## 4. Discussion

We found that repeated QNP treatment caused a robust, reliable, and complex idiosyncratic pattern of stereotypical checking. Exploration activated IEG expression compared to home cage controls in the hippocampus and OFC. Repeated QNP treatment was associated with a lower IEG expression in the hippocampus, suggesting that hippocampal long-term plasticity is compromised in QNP treated rats during stereotypical checking. We found no statistical differences in IEG expression in other brain regions (ACC and mPFC) between QNP treated rats and control rats. To our knowledge, this is the first study to examine plasticity-inducing neural activity in the QNP sensitization model of stereotypical checking.

All QNP treated rats displayed unique and complex checking patterns that were well preserved across experimental days. In contrast, control rats did not develop stable behavioral patterns through experimental days. Since OCD rituals are more structured and complicated than repeated visits to one place [35] our finding improve the face-validity of the QNP-sensitization model.

Next, we showed that repeated administration of QNP diversely affected IEG expression in selected brain regions. While IEG expression was lower in the CA1 area of the hippocampus, IEGs expression was unaffected in OFC, ACC, and mPFC. In CA1, the number of *Homer1a* positive nuclei was lower by 62%, and the number of *Arc* positive nuclei was lower by 32% compared to controls. IEG expression in OFC, ACC, and mPFC was not significantly different between the arena exploring QNP treated rats and control rats.

It is unclear how D_2_/D_3_ agonist QNP simultaneously disrupts hippocampal plasticity while sparing the plasticity of the cortical regions. It is similarly unclear whether the QNP-induced decrease in hippocampal plasticity corresponds with the decrease in the activity of the hippocampus. Current research on this topic is limited and ambiguous. Servaes et al. showed that the [18F]-FDG uptake in the hippocampus decreased by 19.57% in rats chronically treated with QNP [36]. On the other hand, Carpenter et al. reported that local cerebral glucose utilization in the hippocampus of QNP-treated rats remained unchanged when measured by the [14C]2-deoxyglucose [37]. Neither of the studies assessed checking behavior in QNP treated rats, only an increase in locomotion. This makes it more difficult to compare their results with ours.

Despite lower numbers of IEG expressing neuronal nuclei in the hippocampus of QNP treated rats, we found that similarity scores of QNP treated rats and control rats remained comparable. Similarity score is a measure of overlap between IEG expressing populations corrected for random overlap between populations. Absence of difference between similarity scores between QNP treated and control rats indicates that the general function of the hippocampus of QNP treated rats was unimpaired, while the magnitude of plasticity-related changes was reduced. The plasticity-related activity of the hippocampus during stereotypical checking might be decreased but not entirely diminished. The decrease of IEG expression does not mean that all hippocampal processes are compromised. It was found that many active neurons do not express IEGs even in actively behaving animals [38]. It is possible that only IEG dependent processes, such as long-term memory encoding, may be impaired in QNP treated rats. Difficulty in memory encoding was reported in OCD patients with checking compulsions [39]. Interestingly, the confidence in checking-related memories further decreases in both OCD patients and healthy participants who underwent experimental repeated checking protocol [40].

We found that the expression of *Homer1a* and *Arc* in OFC, ACC, and mPFC remained comparable in QNP treated and control rats. Similarly, to our results, studies using MicroPET/CT and local cerebral glucose utilization using the [14C]2-deoxyglucose (2-DG) did not detect statistically significant changes in OFC, ACC or mPFC following QNP treatment [40,41]. In contrast, Asaoka et al. reported that OFC was hyperactive in QNP treated rats using electrophysiological recordings [41]. However, plasticity-related IEG expression may not directly correspond to changes in electrophysiological hyperactivity.

Our study has two major limitations. First, the methods we used did not allow us to reveal the direction of causality between the decrease in the plasticity of CA1 neurons and stereotypical checking. Second, because we did not include acute groups (and we did not assess IEG expression any sooner than on experimental day 11), we are unable to say how quickly the disruption of the plasticity of CA1 occurs following QNP treatment.

Previously we have shown that QNP treatment induces a functional hippocampal deficit reflected by the impairment of reversal learning in the hippocampus-dependent Carousel maze task [26,27]. Present findings of reduction of CA1 plasticity-inducing activity supports the notion that QNP treatment induces functional changes to the hippocampus. We hypothesize that impairment of the hippocampus can lead to an overreliance on habitual circuits (such as CSTC circuits) as a compensation for impaired memory encoding. The notion of the impaired hippocampus as a driver of compulsive behavior is supported by studies showing that damage to the hippocampus is associated with the emergence of compulsive behaviors [21,22,23,24]. Further research is needed to validate hippocampal involvement in OCD.

## 5. Conclusions

In the present study, we show that stereotypical checking—a hallmark of OCD—is associated with impaired plasticity of the hippocampus. Namely, we found that the expression of IEG *Homer1a* and *Arc* is reduced in CA1, the main output region of the hippocampus, during stereotypical checking.

## Figures and Tables

**Figure 1 biomolecules-11-00084-f001:**
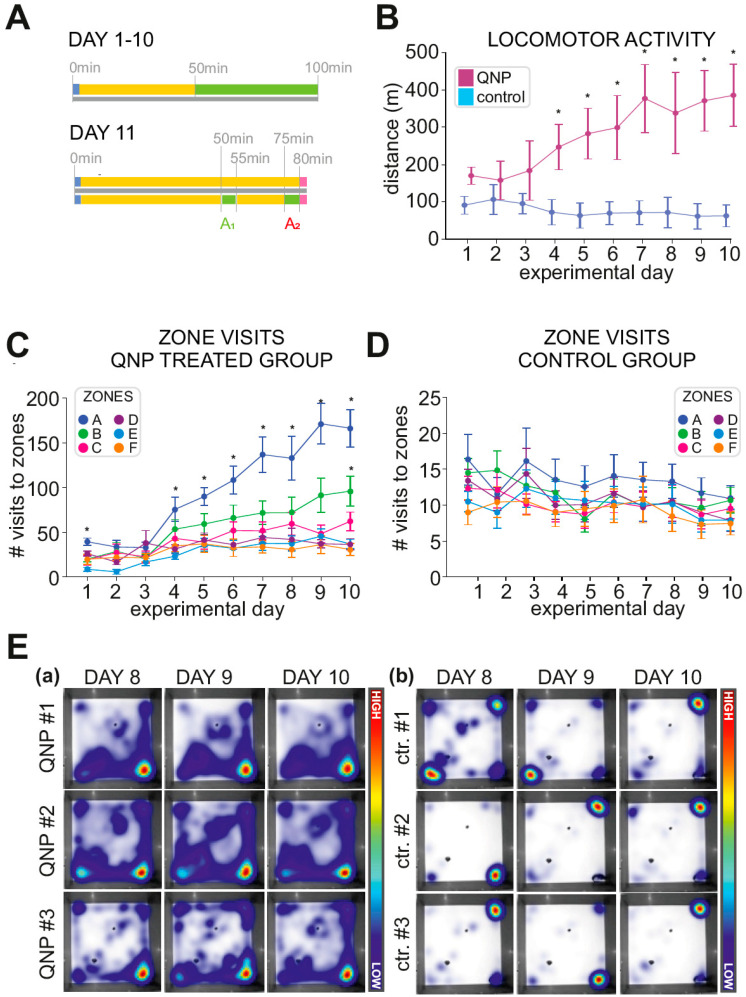
Stereotyped checking in rats repeatedly treated with quinpirole. (**A**) Experimental scheme: During experimental days 1–10 rats received an injection of quinpirole (QNP; 0.5 mg/kg; *n* = 15) or saline (1 mL/kg; *n* = 14) 50 min prior to arena exposure. Rats were then placed into the arena and left to explore for 50 min. On experimental day 11 rats were injected by QNP or saline. After 50 min rats were placed into the arena for 5 min (A_1_), and returned back to the home cage. After 20 min, rats were placed into the arena for the second 5 min exposure (A_2_). Immediately after, rats were sacrificed. 5 QNP-treated rats and 5 saline-treated control rats were kept in the home cage during sessions A_1_ and A_2_ to determine baseline immediate early gene (IEG) expression. (**B**) Locomotor activity of quinpirole (QNP) treated and control rats on experimental days 1–10. QNP treated rats elapsed significantly longer distances than control rats (*p* < 0.001) on experimental days 4–10. (**C**) Visits to zones A–F by QNP treated rats. Zone A denotes the most visited zone, while F denotes the least visited zone from all visits to zones pooled together. QNP treated rats visited zone A significantly more than zones B-F on experimental days 1 and 4–10 (*p* < 0.001) and visited zone B significantly more than zones C–F on day 10 (*p* < 0.05). (**D**) Visits to zones A–F by control rats. Control rats did not visit any zone significantly more than any other. (**E**) Heat-maps illustrate that individual quinpirole (QNP) treated rats had very similar behavioral patterns between experimental days 8, 9, and 10, while locations visited by individual control rats varied between experimental days 8, 9, and 10. Panel (**a**) shows locations visited by three sample QNP treated rats in the last three days of treatment (days 8–10). Panel (**b**) shows locations visited by three sample control rats.

**Figure 2 biomolecules-11-00084-f002:**
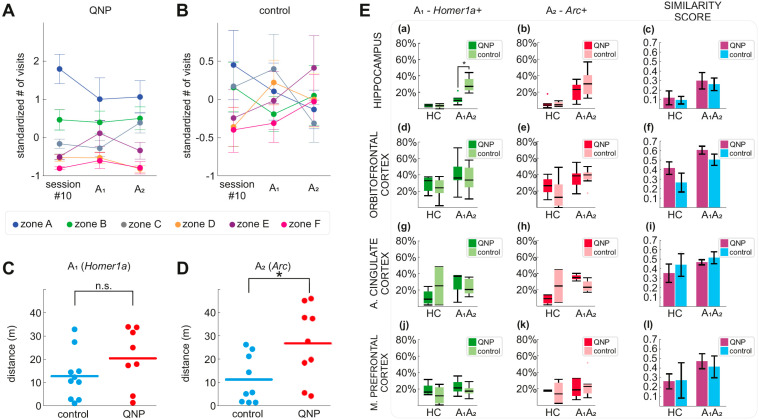
Zone visits, locomotor activity and IEG expression in HPC, OFC, ACC and mPFC during sessions A_1_ and A_2_. Standardized values of visits to zones in quinpirole (QNP) treated (**A**) and control rats (**B**). (**C**) Locomotor activity in QNP treated rats (*n* = 9) and control rats (*n* = 9) in session A_1_ were not significantly different. (**D**) In session A_2_, QNP treated rats were significantly more active than control rats (*p* = 0.025). (**E**) Immediate early genes (IEG) expression and similarity scores of IEG expressing neurons in the hippocampus (**a**–**c**), the orbitofrontal cortex (**d**–**f**), the anterior cingulate cortex (**g**–**i**) and the medial prefrontal cortex (**j**–**l**). Panels (**a**) and (**b**) show that the expression of Homer1a and Arc was significantly lower in quinpirole (QNP) treated rats than in control rats exploring the arena (A_1_A_2_) (*p* < 0.05) in the hippocampus. There were no significant differences in IEG expressions or in similarity scores for treatment and environment in any other brain region. HC denotes home cage groups (QNP: *n* = 6); control: *n* = 5). The asterisk denotes significant between-group differences at *p* < 0.05.

**Figure 3 biomolecules-11-00084-f003:**
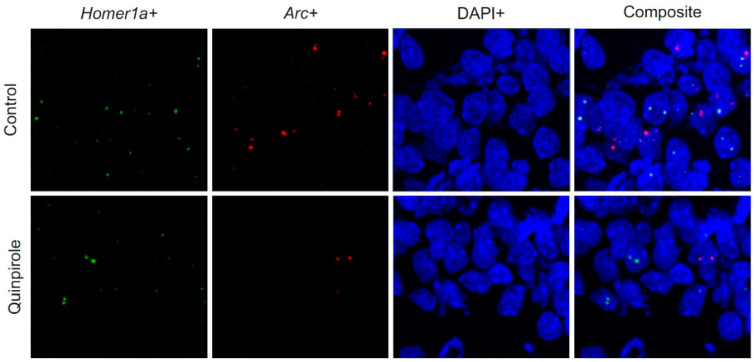
Representative images of immediate early gene (IEG) expression in CA1 area of hippocampus.

## Data Availability

All data generated and analyzed during this study are included in this article. Numeric source data for the presented figures are available from the corresponding author on reasonable request at the desired level of analysis.

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
