# Peer review of "Plasticity-Related Activity in the Hippocampus, Anterior Cingulate, Orbitofrontal, and Prefrontal Cortex Following a Repeated Treatment with D2/D3 Agonist Quinpirole"

_biomolecules, 2021, doi:10.3390/biom11010084_

Round 1

Reviewer 1 Report

The expression of immediate early genes were assessed in different brain regions following quinpirole administration. Authors describes a decrease of IEG-positive nuclei in the hippocampus of treated animals. Authors claimed that quinpirole affects hippocampal plasticity. The manuscript is well written and methodologically correct. The introduction is clear and includes antecedents that support the objectives of this study. There are a few points that should be attended before publication:

  • Behavioural Procedure: Was the apparatus cleaned between subjects? Please include this information in the manuscript
  • How the arena was distributed in 6 zones or areas? What was the zone occupied by the metal objects? Did the visit to the objects differed between groups? Areas of study were defined prior or after the experiment? This information is relevant and should be added to the manuscript.
  • 1. Legend. Authors described that QNP treated rats preferred specific zones of the arena (visited more some zones than other) and this was not the case in controls. Nevertheless, the next sentence describe how the distribution of behaviour on days 8-10 was very similar in QNP treated rats whereas locations visited by controls changed on days 8-10. How is that? It is not contra-intuitive?
  • Regarding IEG expression, why selecting only CA1 and not assessing IEG on other hippocampal areas?
  • Other brain regions involved in OCD did not show a change in IEG expression. This point requires a deeper discussion

Author Response

Response to Reviewer 1

Point 1: Behavioural Procedure: Was the apparatus cleaned between subjects? Please include this information in the manuscript

Response 1: Yes, the apparatus was cleaned between subjects. We added this information to the manuscript (line146-147)

Point 2: How the arena was distributed in 6 zones or areas? What was the zone occupied by the metal objects? Did the visit to the objects differed between groups? Areas of study were defined prior or after the experiment? This information is relevant and should be added to the manuscript.

Response 2: Thank you for the questions. You helped us realize that description of zones was insufficient and needs to be clarified in the text. We added the following section to methods (lines 168-180):

Determination of the optimal location size was important, as entry into the location should be counted only when an animal enters the location and interacts with the object/corner. On the other hand, location entry should not be counted when a rat is merely passing by. To determine the optimal size and shape of the location containing the objects and the cornets, we used an empirical approach. To that aim we utilized data from pilot experiments. The optimal location was either circular (objects) or quadri-circular (corners). The radius of locations was determined by comparing the scoring of the software with hand made scores by an experimenter blinded to the treatment condition. We selected a final radius of each location so that experimenter-determined counts of the visit to objects/corners corresponded exactly to the counts of visits by the Bioobserve software. In our setup, object zones had a radius of 10 cm with the object in the center, while quadri-circular corner zones had a radius of 15 cm with the tip in the corner of the arena. Of course, the radius of the zones would be different if objects were of different sizes or if animals were bigger/smaller. The illustration of zones is in Figure 1A.

Point 3: 1. Legend. Authors described that QNP treated rats preferred specific zones of the arena (visited more some zones than other) and this was not the case in controls. Nevertheless, the next sentence describe how the distribution of behaviour on days 8-10 was very similar in QNP treated rats whereas locations visited by controls changed on days 8-10. How is that? It is not contra-intuitive?

Response 3: We did not completely understand the question. However, we made a minor text change that will hopefully clarify the test (lines 283-284).

Point 4: Regarding IEG expression, why selecting only CA1 and not assessing IEG on other hippocampal areas?

Response 4:  Thank you for the question. Text in italics was added to the manuscript (lines 97-118):

ACC and OFC are implicated in OCD pathophysiology. CA1 area was selected because most of the IEG expression occurs in CA1 area: approximately 30-40% of CA1 neurons express the IEG Arc after behavioral exploration of a (novel) environment. In comparison, only 10-15% of CA3 neurons and 2% dentate granule cells express Arc in the same conditions. Therefore, the majority of the active neurons in the hippocampus are from CA1. For this reason, most studies on IEG expression in the hippocampus focus on CA1. Furthermore, expression of Arc in the DG is not short-lasting such as in CA1, CA3, and all other brain regions, but once triggered, remains high for as long as 8 hours.

Point 5: Other brain regions involved in OCD did not show a change in IEG expression. This point requires a deeper discussion.

Response 5: Thank you for this suggestion. However, we restrained from discussing the topic, as it would be mere speculation at this point. The conflict is referred to in this sentence “It is unclear how D2/D3 agonist QNP simultaneously disrupts hippocampal plasticity while sparing the plasticity of the cortical regions” (lines 397-398), further, lack of effect in cortical regions is discussed on lines 435-441.

Reviewer 2 Report

In this paper the authors evaluated the effects of QNP-induced checking behaviour on IEG expression in various regions of the brain. I do have some concerns about the novelty of the work. The QNP model of checking is well established. Thus the only real novel data comes from the IEG studies, within which only the hippocampus showed a change. Even if the recommended studies below are performed (required for model validation), this will not add to the novelty of the paper.

Are there any animal studies looking at OCD-like behavior in rodents and the role of the hippocampus (in addition to authors own work)? These should be included in the intro.

Why were animals left for 50 minutes in their home cages post injection? This does not appear to be in line with the Szechtman protocol which placed the animals in the open field immediately post injection.

Zones A-F were assigned according to frequency of visits. Were zones A and B ones that had the object?

There are several parameters that are missing that are necessary to validate the model, especially given the modifications from the original Szechtman protocol (# objects, timing of open field exposure, size of open field). For example, path stereotypy, frequency of checking, recurrence time of checking (return time), length of check, relative focus on key place (% of total stops). See Perreault et al., 2007.

Representative images are required for the IEG expression.

Fig. 2B – The mean line for the QNP rats look like it should be much lower.

Sample sizes should be included in the figure captions.

Author Response

Response to Reviewer 2

Point 1: In this paper the authors evaluated the effects of QNP-induced checking behaviour on IEG expression in various regions of the brain. I do have some concerns about the novelty of the work. The QNP model of checking is well established. Thus the only real novel data comes from the IEG studies, within which only the hippocampus showed a change. Even if the recommended studies below are performed (required for model validation), this will not add to the novelty of the paper.

Response 1: We agree; we too think that only IEG data are novel. However, we think it is advantageous that IEG expression data are reported on the background of a well known animal model: thus, they can be much easily put into the context of previous work.

Point 2: Are there any animal studies looking at OCD-like behavior in rodents and the role of the hippocampus (in addition to authors own work)? These should be included in the intro.

Response 2: Thank you for the reminder. To our best knowledge, no other studies are looking at the role of the hippocampus in the etiology of OCD-like behavior.  We added this sentence to the end of the introduction on lines 88-89.

Point 3: Why were animals left for 50 minutes in their home cages post-injection? This does not appear to be in line with the Szechtman protocol which placed the animals in the open field immediately post injection.

Response 3: We decided to deviate slightly from the Szechtman protocol for two reasons. One is the biphasic effect of quinpirole on behavior and the second is efficiency.

First, quinpirole induces a period of hypo-locomotion followed by hyper-locomotion. This biphasic effect was attributed to activation of presynaptic D2 receptors (sedation) and postsynaptic D2 receptors (hyperlocomotion) (Eilam & Szechtman, 1989, DOI: 10.1016/0014-2999(89)90837-6). Empirically, we found that hyperactivation has an onset at around 50 min after the first QNP injection (although this period shortens with subsequent injections).

Second, in the FISH experiment, we use 4 different treatment groups. If we run 2h long sessions, we would have to comply with n=1 for each group; or n=2 if we did 16h work a day. However, 16 hours workday cannot be fitted into the 12h light phase that we conduct experiments in. Since we found that strong stereotypic behavior is present using the slightly modified protocol as well, we felt confident to proceed. Potentially, as this modification produces similar results with half of the invested time, the QNP sensitization can become a more efficient model of stereotypical behavior.

Point 4: Zones A-F were assigned according to the frequency of visits. Were zones A and B ones that had the object?

Response 4: That is a very interesting question. Moreover, there were differences in the identity of the most frequented zones between the groups. In QNP treated rats the most visited zone, zone A, was most often a zone containing a corner; similarly, in controls zone A was also most frequently a zone containing a corner. On the other hand, in QNP treated rats the second most frequented zone, Zone B, was a zone containing an object; while in control rats it was most often zone containing a corner. To summarize, QNP treated rat stereotypic activity comprised most often of movement between the selected corner and the selected object, while control animal locomoted between corners of the arena.

As this may be of interest to other readers too, we added text in italics to the results section of the manuscript (lines 324-330).

Point 5: There are several parameters that are missing that are necessary to validate the model, especially given the modifications from the original Szechtman protocol (# objects, timing of open field exposure, size of open field). For example, path stereotypy, frequency of checking, recurrence time of checking (return time), length of check, relative focus on key place (% of total stops). See Perreault et al., 2007.

Response 5: Yes, we deviated from the original protocol; however, all the deviations were approved by prof. Szechtman via email correspondence. It was professor Szechtman who suggested that on a smaller arena we should place only one or two objects. Of course, we wish that arena could be bigger, but we do not have an experimental room that would fit a 2x2m arena. Yet, we believe that showing that such modification still produces solid stereotypical behavior points to the robustness of the observed behavioral phenomena.

To clarify the zone selection for the reader, we added the following to the manuscript (lines 168-180):

Determination of the optimal location size was important, as entry into the location should be counted only when an animal enters the location and interacts with the object/corner. On the other hand, location entry should not be counted when the rat is merely passing by. To determine the optimal size and shape of the location containing the objects and the cornets, we used an empirical approach. To that aim, we utilized data from pilot experiments. The optimal location was either circular (objects) or quadri-circular (corners). The radius of locations was determined by comparing the scoring of the software with hand made scores by an experimenter blinded to the treatment condition. We selected a final radius of each location so that experimenter-determined counts of the visits to objects/corners corresponded exactly to the counts of visits by the Bioobserve software. In our setup, object zones had a radius of 10 cm with the object in the center, while quadri-circular corner zones had a radius of 15 cm with the tip in the corner of the arena. Of course, the radius of the zones would be different if objects were of different sizes or if animals were bigger/smaller. The illustration of zones is in Figure 1A.

Point 6: Representative images are required for the IEG expression.

Response 6: representative images of IEG expression were added as new Figure 3

Point 7:  Fig. 2B – The mean line for the QNP rats look like it should be much lower.

Response 7: We apologize; there was some distortion of the images (we noticed it in other Figures as well). For some reason, data points are missing from Figure 2B. These data points were present before the picture was uploaded. We will make sure that all figures are correct if the present manuscript is published.

Point 8: Sample sizes should be included in the figure captions.

Response 8: Thank you for the suggestion: we added sample size to both figure legends (lines 269-270 for figure 1 and lines 292 and 300 for figure 2)

Round 2

Reviewer 1 Report

Authors answered all my queries.

Reviewer 2 Report

The authors have addressed my concerns.